# Patient-Reported Perception of Exercise and Receptiveness to Mobile Technology in Cancer Survivors Living in Rural and Remote Areas

**DOI:** 10.3390/curroncol32020067

**Published:** 2025-01-27

**Authors:** Myriam Filion, Saunjoo L. Yoon, Becky Franks, Dea’vion Godfrey, Carina McClean, Jackson Bespalec, Erin Maslowski, Diana J. Wilkie, Anna L. Schwartz

**Affiliations:** 1Faculty of Kinesiology, Sport, and Recreation, College of Health Sciences, University of Alberta, Edmonton, AB T6G 2R3, Canada; mfilion@ualberta.ca; 2Department of Biobehavioral Nursing Science, Center for Palliative Care Research and Education, College of Nursing, University of Florida, 1225 Center Drive, Gainesville, FL 32610, USA; diwilkie@ufl.edu; 3Cancer Support Community Montana, Bozeman, MT 59715, USA; becky@cancersupportmontana.org; 4College of Medicine, University of Florida, Gainesville, FL 32611, USA; dgodfrey1@ufl.edu (D.G.); carina.mcclean@ufl.edu (C.M.); 5Kearney Division, College of Nursing, University of Nebraska Medical Center, Kearney, NE 68198, USA; jabespalec@unmc.edu; 6College of Letters and Science, Montana State University, Bozeman, MT 59715, USA; erinkatemaz@gmail.com; 7Coleman Health LLC, Parks, AZ 86018, USA; annalschwartz@icloud.com

**Keywords:** exercise, fatigue, survivorship, digital technology, focus group, distance-based intervention, rural area, frontier states, health disparities, cancer

## Abstract

Purpose: Cancer survivors in rural and underserved areas face barriers such as limited access to oncology exercise programs and limited facilities, contributing to health inequities in cancer survivorship. This study explored cancer survivors’ thoughts on exercise and mobile technology for exercising with a mobile application (app) during and after treatment in rural and remote areas. Methods: Three online focus groups were conducted in February 2024 using semi-structured interviews with 12 open-ended questions. Eligible participants were adult cancer survivors or caregivers living in medically underserved areas, English-speaking, consented to being audiotaped, and attended one 60-min group interview. The discussions were transcribed verbatim and analyzed via a content analysis approach with consensus. Results: Fifteen participants attended from four States. None of the participants were advised to exercise; availability of exercise resources depended on geographic location and a cancer-specific exercise app was desired. They understood the benefits of exercise after diagnosis but expressed a need for more guidance during treatment. Geographic location shaped their activities, with most engaging in daily physical tasks rather than structured exercise. Most participants were receptive to using an exercise app to manage fatigue. Suggested key features to exercise with an app included live trainers, exercise checklists, visual benchmarks, and programs tailored to different fitness levels. Conclusions: These results emphasize the need for personalized resources, guidance, and on-demand accessibility to an exercise oncology app. A cancer-specific exercise mobile app will mitigate health inequities for cancer survivors residing in rural and remote areas.

## 1. Introduction

It is estimated that 18.1 million cancer survivors will be living in the United States (U.S.) in 2024. The number will increase by 24.4% to 22.5 million in 2032 [1]. Exercise improves cancer-related symptom burdens and reduces postoperative complications and length of hospital stay among cancer survivors [2,3,4]. Further, exercise significantly impacts disease-free survival, including a 30–60% reduction in recurrence and mortality of colorectal and breast cancers [5,6,7,8]. However, only 14.2% of cancer survivors meet the physical activity recommendations [2,9,10,11], with particular concerns about cancer survivors in rural and underserved areas. The most consistent barriers to exercise among cancer survivors are cancer and treatment-related symptoms (e.g., cancer-related fatigue), lack of healthcare provider knowledge in oncology, guidance regarding safety in cancer survivor exercise parameters, and access to oncology exercise programs or an exercise oncology trainer [12]. It is unknown if survivors in rural areas have additional barriers to exercise.

People in rural and medically underserved areas experience significantly greater cancer health disparities in cancer incidence and outcomes (e.g., survival rate) [13], including a 2.7% higher incidence and 9.6% higher mortality (*p* < 0.001) than in urban areas [14]. They experience a higher burden of disease and are more likely to be inactive after a cancer diagnosis. The more rural the area, according to the Rural–Urban Continuum Code, the higher the level of symptom intensity [15,16,17]. They face difficulties accessing oncology care, limited oncology services, financial issues, access to clinical trials, and geographic and transportation barriers [18].

In addition to a higher disease burden among cancer survivors in rural and remote areas, many social determinants of health impact their survivorship. For instance, Arizona is classified as a rural and frontier state (population density of six or fewer residents per square mile) with 99,399 square miles of frontier land (frontier counties). An insufficient healthcare workforce, remoteness, geographic isolation, and cost of travel impede access to quality and timely healthcare [19]. Rural, Native American, and underserved people have a higher burden of disease because of geographic disparities (distances to health care) and physical inactivity [20,21]. In rural Arizona, residents tend to be older (22.6% over 65 years old compared to 16.5%) than in urban areas. Sixty-nine percent of older adults in rural areas live below the poverty level [22,23]. Florida has 32 out of 67 counties that are defined as rural areas [24], and 27 of the 32 rural counties are located in North Florida. These 27 counties are all classified as Medically Underserved Areas (MUA) and/or MUA/low income (MUP) using the Health Resources Service Administration’s (HRSA) definition of Rural [25]. Montana is a state of 145,552 square miles with three urban areas: Billings, Missoula, and Great Falls [26]. It has a population density of 6.86 people per square mile, with 33% of the population living in rural or frontier areas. All of Montana is designated by HRSA as medically underserved, with six counties classified as MUP [27]. Whereas the population is 87% White, with 6.07% Native American and 4.49% two or more races, it has an underrepresented population of rural and MUA/MUP cancer survivors who cope with harsh weather, extreme distances, limited access to care, and 12.5% of the population (17% of children) living 100% below the federal poverty level. Rural Montana families face challenges, including distance to medical care, limited specialists and healthcare facility locations, and access to supplemental services. North Dakota is a rural state and the 4th least populated state in the U.S., with about 780,000 residents. Thirty-six (36) out of 53 counties in North Dakota are ‘frontier’ counties. Native Americans compose the largest group (5.6%) among minority populations in North Dakota [28].

These figures underscore the unique opportunity for cancer survivors in these four states to access exercise remotely and receive benefits. Digital technology, or the use of a mobile application (app) to provide cancer-specific exercise prescriptions, may be part of the solution to tackle health disparities and improve the health and well-being of cancer survivors in rural and underserved areas [29,30]. In the general adult population, mobile apps have effectively increased exercise levels and are widely used for promoting exercise [31,32,33]. Similarly, apps have demonstrated effectiveness in symptomatic populations, including individuals with cardiovascular disease, obesity, and during menopause [34,35]. In oncology patients, distance-based exercise interventions, including mobile apps, have shown promise in increasing exercise levels and reducing fatigue, with additional benefits such as improved aerobic capacity, upper body function, muscle strength, and quality of life [36,37]. However, access to and use of these tools among oncology patients can vary widely. Barriers such as limited internet access in rural areas and lack of familiarity with technology among older adults may restrict some patients’ ability to adopt these interventions [33]. Understanding how different groups of cancer survivors access and use exercise apps is critical to ensuring these tools are inclusive and effective. By addressing these disparities, future interventions can be better tailored to meet the diverse needs of the cancer population.

Since 2023, the Cancer Exercise mobile app has been freely available on the App and Google Play Stores [38]. Providing individual exercise prescriptions, the Cancer Exercise app makes exercise available to adult cancer survivors of all ages, levels of physical condition, types, stages of cancer, and types of treatment. Whether new to fitness or long-term exercisers, the Cancer Exercise app guides cancer survivors through every step of exercise to make exercise part of daily life. This app continues to evolve through an iterative process, integrating valuable feedback from cancer survivors to better meet their needs and enhance their experience. The purpose of this focus group study was to (1) understand cancer survivors’ and caregivers’ thoughts and preferences about exercise and mobile technology and (2) assess receptiveness to a personalized cancer-specific exercise mobile app during and following cancer treatments in rural, remote and underrepresented areas of four states, including Arizona, Florida, Montana, and North Dakota.

## 2. Sample and Methods

### 2.1. Study Design

This study employed a qualitative approach, using focus groups and semi-structured interviews to explore perspectives on exercise needs and evaluate exercise programs specifically designed for cancer survivors (Figure 1). Three focus group sessions were conducted online using a cloud-based video conferencing platform, to connect to participants living in the rural and remote areas of four states: Arizona, Florida, Montana, and North Dakota.

The sample was selected using snowball, word-of-mouth, and purposive sampling methods from rural areas of Arizona, Montana, Florida, and North Dakota. Eligible criteria were cancer survivors in any stage and their caregivers who live in designated rural, remote, and/or underserved areas of these four states. The eligible participants were identified by the Cancer Support Community in Montana, an organization in Florida affiliated with the Cancer Support Community, and cancer clinics in Arizona. Eligible participants had to be adult cancer survivors or caregivers who (1) lived in one of four states (Arizona, Florida, Montana, or North Dakota), (2) could communicate in English, (3) were able to attend up to 60 min of group interviews remotely, and (4) agreed to be audiotaped.

### 2.2. Ethical Aspects

This study was approved as exempt by the Institutional Review Board (IRB) of the University of Florida, Protocol# ET00022194, in January 2024. One research team member (B.F.) conducted the focus group discussions. Each session began with the project’s purpose, participant expectations, the right to withdraw at any time during the interview, ground rules, and a clarification that the session would last less than 60 min.

### 2.3. Data Collection

The research team developed the ground rules and 12 open-ended questions, focusing on exercise experience during and after cancer treatments and the willingness to use an exercise mobile application (app), which can be individualized. The questions were crafted to align with the Cancer Exercise app’s planned components and development trajectory, ensuring we could gather relevant insights to inform the next stages of refinement. Examples of the questions are “*What exercise did you do before cancer? Did you try to exercise during your cancer treatment? What made it hard to exercise during and after cancer treatment? What would have helped make exercise easier?”* “*If you knew your exercise would improve your fatigue, do you think you would use the app? Do you have internet access to use an app?” and “How would you feel about using the app video made for someone like you to help you get started and stay on an app-based exercise program?*” However, we did not present any specific app and the participants were blinded to the Cancer Exercise app that was discussed previously. We wanted to avoid influencing participants’ responses based on the app’s current state. This approach allowed us to capture various participants’ preferences, needs, and innovative ideas without limiting feedback to the existing design.

For all three focus groups, the interviewer asked the same questions. After the IRB approval, research team members who worked with cancer survivors or knew survivors in designated regions of four states contacted eligible participants. When the eligible participants agreed to be contacted, their contact information was shared with one research team member. After obtaining the contact information with permission to consent to the study participation, the research team emailed the eligible participants the consenting information, study description, study procedures, confidentiality, and potential dates and times. Before the scheduled interview date, each participant received an email reminder and the Zoom link with instructions on accessing the meeting. After completing the interview, research team members emailed all participants to show appreciation and sent them a $25.00 gift card as a small token. All three focus group interviews were audiotaped for analysis.

### 2.4. Data Analysis

Data from the focus group were analyzed using principles of the Krueger approach [39], which outlines a clear series of steps for interpreting the data [40]. The first step involved producing a verbatim transcript of the discussions from all three focus groups, completed by three authors (M.F., D.G., C.M.) to ensure internal validity across the data. The full transcript was then reviewed collaboratively to ensure nothing had been missed in the discussions. Data analysis occurred on two levels [41]. At the first level, the data were categorized into major themes, emphasizing reporting what was said without making assumptions. The second level focused on identifying minor themes, which involved interpreting the major themes and understanding how they emerged, ultimately leading to a consensus across the data.

## 3. Results

### 3.1. Demographic Characteristics

The focus group interview was conducted in February 2024. A total of 15 participants (eight women and seven men), including nine cancer survivors and six caregivers of the cancer survivors, attended the three focus group interviews. Of those, five participants were from Arizona, four were from Florida, four from Montana, and two were from North Dakota. Table 1 depicts the characteristics of the study participants.

### 3.2. Experiences of Exercise During and After Cancer Treatment

The analysis process resulted in five key themes: (1) exercise resource availability, (2) exercise guidance, (3) impact of inactivity perception, (4) exercise behavior and barriers, and (5) exercising with a partner (Table 2).

#### 3.2.1. Exercise Resource Availability

During their cancer care, none of the participants had been informed about specific exercise programs or resources available in their area to stay active during treatments. The availability of exercise options depends largely on location, which influences both the types of activities in which people can engage and the resources they can access. “*I live in the middle of nowhere so it’s hard for me to get any type of support*”. Walking, being accessible to most, was commonly mentioned as an exercise when participants felt physically able. Physical therapy (PT) was typically pursued only when prescribed by a healthcare provider. “*My husband had a prescription for PT, so we do have PT in our rural community, and he did that after his first cancer […]*”.

#### 3.2.2. Exercise Guidance

Overall, none of the participants were explicitly advised to exercise. When exercise was mentioned, it was nonspecific and usually involved walking. The recommendations they received were general, encouraging them to do as much as they could, maintain their current activity levels, and walk for a certain amount of time. One participant expressed fear about exercising during cancer treatment and sought advice from her doctor, who recommended monitoring for signs like shortness of breath and a rapid heartbeat. Another participant mentioned: “*I took exercise upon myself, I took walks, and I have a basketball court near my place to play. My wife gave me a lot of work to do when I first got home. I have a garden, and I look at my garden. I took a whole building down, overtop shade, and I have replaced it. So, I’ve gotten a lot of exercise*”.

#### 3.2.3. Impact of Inactivity Perception

All focus group participants had some prior exercise experience before their cancer diagnosis, which made them aware of the benefits of staying active during treatment. They understood that inactivity could worsen fatigue during treatment, and their pre-existing knowledge of the benefits of exercise motivated them to remain active. One participant noted: “*It’s common sense that moving more makes you feel better. If they had told me then, I would agree, but I’m not sure it would have changed anything. Maybe if it came from a doctor, I would have been more inspired to take it seriously*”. However, participants expressed a need for clearer guidance on how much exercise would be beneficial. One participant shared an experience: “*I check my blood values every time I undergo immunotherapy. I see some levels, like my immune markers, at the desired range, and I wonder if exercise is helping or if I’m overdoing it—especially with my red blood cell count. I would appreciate more guidance on whether I’m doing too much or too little. I do about two hours of active exercise daily, a mix of hiking, biking, and yoga, but I have no idea if that’s too much for my condition. Guidance would be really helpful*”.

#### 3.2.4. Exercise Behavior and Barriers

When it came to their pre-cancer exercise routines, most participants tried to continue their activities despite undergoing treatment. Many were farmers or primary caregivers for young children, which kept them physically active before and after their diagnosis. As one participant shared: “*The first time I had cancer, I had two high school-aged kids who were involved in everything. We never stopped chasing cattle or doing what we had to do—that was our life. This time, we’ve retired from farming and sold the cattle. Now we’re busy chasing six grandkids*”. Others were former athletes, although some did not follow a specific exercise routine before the diagnosis. The athletes reported being more in tune with their bodies, and many participants noted a significant decrease in strength following their diagnosis. Although many attempted to stay active during treatment, this attempt often involved daily tasks rather than intentional exercise. One participant explained: “*Before cancer, I was very active, always doing physical work since I live in a remote area. I made it through treatment, but at the hospital, all I could do was walk the halls. When I got back home, I had no strength and had to work to build it back up. I’m in the process of rebuilding myself, with help from my kids*”.

#### 3.2.5. Exercising with a Partner

Participants expressed a strong preference for exercising with a partner, emphasizing the importance of social interaction and motivation in maintaining exercise behavior. Most participants mentioned that having a partner would help with accountability. One participant noted: “*In the past, I’ve found that accountability is a good motivator. Being accountable to someone else, or having someone who shows up for you, makes a big difference*”. However, a challenge arises for those who don’t have an exercise partner. A participant shared: “*I don’t have an accountability partner for exercise, but I’m sure it would be motivating. If someone asked me to go for a walk, I’d go—I couldn’t come up with an excuse. An accountability partner would be motivating and good company*”. For those who do have a partner, family members often fill that role. A caregiver participant explained: “*Yes, if I didn’t bring it up, he [the husband] might not think about exercising as often. Being a partner team helps us stay consistent*”.

### 3.3. Receptiveness to the Use of a Personalized Cancer-Specific Exercise Mobile App

In the second part of the focus group discussion, we aimed to assess the participants’ willingness to use digital technology and identify the factors that would facilitate its use. The data analysis identified five key themes: (1) Internet accessibility for app usage, (2) exercising with a partner via an app, (3) role of videos in exercising with a partner, (4) role of a trainer in exercising with a partner, and (5) motivation to exercise (Table 2).

#### 3.3.1. Internet Accessibility for App Usage

Internet access for using digital technology did not appear to be a concern, as all participants reported having access. Most participants were receptive to using an exercise mobile app, particularly if it could help manage fatigue. “*Yes, I would use an app. I would find it interesting if it catered to what you were able to do, allowed to do, or the restrictions. So, an exercise app definitely would be something we would look at and use. We have access to the Internet*”. However, a few participants who were less inclined to use an app mentioned that they do not regularly use their phones. A participant explained: “*I don’t think it would be helpful because my phone stays in the car, and I rarely charge it. And we’re already active, so I’m not sure it would make much of a difference*”.

#### 3.3.2. Exercising with a Partner via an App

Participants’ perceptions of exercising with a partner varied in the context of individualized exercise prescriptions. Having partners at different stages of life or health can enable many participants to work out together or continue their exercise routines alongside someone else. However, the enjoyment of exercising with a partner can vary, and some participants feel it may be more challenging than beneficial. One participant suggested that making the exercises into a game or incorporating a group feature could enhance the experience. “*We are in different stages of life. She’s probably in better shape than I am. But it would depend on the situation if we have the same capabilities and restrictions taken into consideration. But the option to have a group or if we sign up for the same challenge and we could see each other’s progress that would be fun to make a game of it. Or maybe attend one of the live sessions together, but it may not work out because of logistics*”.

#### 3.3.3. Role of Videos in Exercising with a Partner

Participants shared their thoughts on using personalized videos to help them engage with and stay committed to an app-based exercise program. Overall, the idea of videos was well-received, particularly by those unfamiliar with recommended exercises. Participants felt that videos would be helpful, provided they were engaging. One participant shared: “*I love videos. I think they’re easier to follow than reading. Following along would be more encouraging and dynamic than just having written instructions*”. They also noted that videos would benefit visual learners and be especially useful for demonstrating proper form and positioning. Another participant specified: “*I agree, especially for something like yoga or Pilates, where I’m not familiar with the correct positioning for poses. A video would definitely help in that regard*”. Participants also discussed how videos could show ways to exercise with a partner, even if partners have different fitness levels or different needs. They suggested that partner videos should include both beginner and advanced options. Clear demonstrations of what to do would be especially helpful for those with varying fitness levels. One participant recommended: “*The implementation will depend on the structure of the class, particularly if there are options for different fitness levels where individuals can engage in more rigorous exercises with the right gear. Otherwise, I don’t believe it would have an impact. As long as there’s clarity about our current fitness levels and the class offers multiple levels, I can progress to the next one when I feel stronger*”.

#### 3.3.4. Role of a Trainer in Exercising with a Partner

The idea of having a live exercise trainer was generally well-received, particularly among participants who were less familiar with the exercise, as they appreciated the immediate, personalized feedback. One participant stated: “*Yes, I think that would be helpful, especially if you have questions or want to ensure you’re doing the exercise correctly and that your form is effective and safe*”. However, several drawbacks were noted. Scheduling conflicts were identified as a likely issue, and cost was mentioned as a potential deterrent. One participant explained: “*I need the flexibility of the time and get signed up on the app whenever it works for me. Setting up with another trainer then you have to work with the schedule and it may or may not work as well. Scheduling has to be easy*”. Additionally, some participants felt that a live trainer might slow them down, indicating that this feature might appeal more to beginners than those with more exercise experience. Participants concluded that having a trainer would make it easier to exercise with a partner, even if they have different exercise prescriptions. They felt that a trainer would help to guide them with the type and intensity of exercises. Participants emphasized the importance of a trainer who can assess individual capabilities and provide guidance on whether pain is a normal part of the workout.

#### 3.3.5. Motivation to Exercise

Participants expressed that maintaining a routine and regaining their pre-diagnosis strengths and abilities are key factors driving their motivation. One participant explained: “*My motivation is probably to get myself back to where I was before cancer—being able to care for my family like I used to. That’s what drives me. It might take a while, but that’s my goal*”. Other motivating factors identified to stay in an exercise program included weight loss and seeing progress over time. A participant mentioned: “*Progress is definitely the motivator—reaching milestones, checking off goals, and seeing how you evolve in the program*”. Some participants also highlighted the importance of having an accountability partner or interacting with others, as it can be harder to stay on track alone. “*I think if you are self-disciplined this works, but otherwise you have to have some level of accountability. Whether it’s an exercise buddy, friend, or someone who can help you stay on task or be in that realm with you*”. Participants also identified key features that would help them stay motivated with an exercise app, such as tracking multiple health metrics, an exercise checklist, visual benchmarks, and programs tailored to different fitness levels.

## 4. Discussion

This qualitative study provides new insights into the exercise experiences of cancer survivors during and after treatment, as well as their receptiveness to a cancer-specific exercise mobile app, particularly in the context of rural and underserved areas. One uniqueness of this study lies in its inclusion of caregivers’ perspectives, offering deeper insights into the role of social support in exercise. Cancer survivors recognize the importance of exercise during and after treatment, with many motivated by the desire to regain pre-diagnosis fitness conditions. Geographic location significantly influences the types of physical activities in which they engage. For most participants, physical activity was already a part of their daily routine, often through traditional forms such as farming or other manual tasks. However, the ability to stay active can be affected by challenges such as fatigue and loss of strength during recovery, which impact physical functioning. Therefore, it comes as no surprise that fatigue is the most common symptom reported by cancer survivors and may be linked to other symptoms, such as muscle weakness [42,43].

Participants from rural areas were interested in adopting a cancer-specific exercise app as part of an intervention, especially if it could help manage fatigue and provide personalized guidance. These findings align with previous studies reporting that digital technology is well accepted by cancer survivors, describing digital technology as flexible, convenient, and easy to use [29,36,44]. Participants indicated they had internet access but also raised concerns about scheduling conflicts, cost, and travel time, all of which can be mitigated using the on-demand mobile app format. These findings also are consistent with previous research that explored the specific needs and barriers facing cancer survivors in rural communities [45].

Our participants expressed the need for ongoing support, such as exercising with a partner or a live trainer, for accountability, consistency, and motivation. These findings support previous studies that show that higher levels of social support enhance engagement in physical activity [46,47]. Similarly, Cooper et al. [48] underscore the importance of practical social support, such as virtual exercise coaching, as a key behavior change technique. To support partner workouts, participants suggested incorporating app features that accommodate different fitness levels within the same exercise program. Additional features like tailored videos, visual benchmarks, and health-tracking metrics were identified to increase motivation to exercise with an app.

The more rural and remote the area, the more limited the access to exercise resources, which leads to health disparities [29,30]. Long travel distances to exercise facilities, which involve time and cost, often limit cancer patients’ participation in supervised exercise programs and reduce referrals from healthcare professionals [49]. However, a cancer-specific mobile app may prove effective in overcoming these barriers, giving survivors the flexibility to incorporate exercise into their daily routines anytime, anywhere. Similar to cancer survivors in other rural communities [29,45], our participants expressed concerns about a lack of familiarity with exercise techniques, safety, and uncertainty about the appropriate exercise volume for their condition. These concerns highlight the need for educational resources and personalized support, areas where a cancer-specific mobile app can offer flexibility, making exercise more accessible and effective for cancer survivors. To promote equitable access for cancer survivors in underserved rural and remote areas, future initiatives should incorporate personalized digital technology solutions that provide social interaction and tailored guidance.

Limitations of this study include possible selection bias. Many cancer survivors are not interested in physical activity and these individuals might not be interested in coming forth and sharing their thoughts about exercise and perspective on using a cancer-specific mobile app. Although the results provide valuable insights into patients’ experiences and interest in digital technology, it is important to recognize that these perspectives may not fully capture the views of all individuals diagnosed with cancer regarding exercise. Another limitation is the lack of generalizability of the findings. As a qualitative study using non-probability sampling, the results may not apply to other cancer survivors in this or other rural countries. Nevertheless, the inclusion of participants from four different states was sufficient to capture the experiences of individuals living in diverse rural and remote communities, enhancing the depth and contextual relevance of the data.

In conclusion, innovative strategies are needed to integrate exercise interventions into rural and remote communities for cancer survivors, and this study highlights the experiences and specific needs of cancer survivors when using digital technology. An app-based exercise program for cancer survivors should be flexible, personalized, and supportive, with features that address the unique needs of individuals at different stages of treatment and recovery. Health equity remains a key concern in cancer care, and this study highlights how digital technology, such as cancer-specific exercise apps, can play a crucial role in reducing disparities by addressing the unique needs of cancer survivors, particularly in rural and underserved areas. Future research should focus on testing cancer-specific exercise mobile apps to evaluate their feasibility, effectiveness, and potential to improve health outcomes for cancer survivors, particularly in rural and underserved areas.

## Figures and Tables

**Figure 1 curroncol-32-00067-f001:**
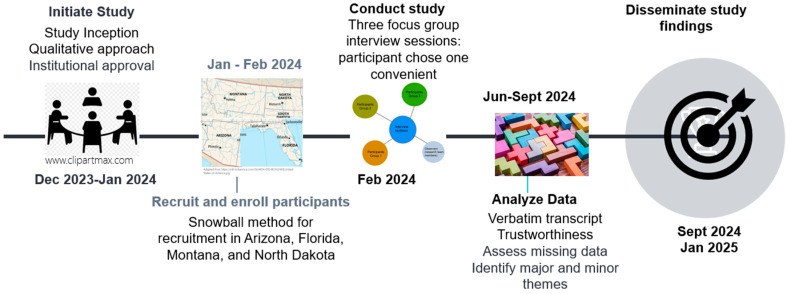
Timeline of Study Activity.

**Table 1 curroncol-32-00067-t001:** Demographic characteristics (N = 15).

Demographic Variables	N (%)
Sex Male Female	7 (47%) 8 (53%)
Participant role Cancer survivor Caregiver	9 (60%) 6 (40%)
Type of cancer (N = 9) Breast cancer Neuroendocrine carcinoma Melanoma Leukemia Papillary carcinoma Multiple myeloma Hodgkin lymphoma	3 (34%) 1 (11%) 1 (11%) 1 (11%) 1 (11%) 1 (11%) 1 (11%)
Participant location Arizona Florida Montana North Dakota	5 (33%) 4 (27%) 4 (27%) 2 (13%)

**Table 2 curroncol-32-00067-t002:** Interview Theme Summary.

Interview Theme	Summary	Sample Quotations
Section 1: Experiences of exercise during and after treatment
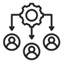	Availability of Resource	None of the participants were informed about exercise programs or specific resources during treatment.	*It’s been a few years since I was on treatment so at the time I wasn’t aware of any specific programs.*
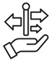	Guidance	None of the participants were explicitly advised to exercise. The recommendations were not specific.	*I don’t recall a directive but they give you a stack of books and resources. I do recall there being pieces in there about being physically active and the importance of that.*
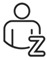	Inactivity Perception	Most participants understood that inactivity could worsen fatigue during treatment and that exercise has benefits.	*I found myself resting more and neuropathy was increasing so it was counterintuitive, I was creating my problem. So I learned the hard way to ignore exercise, which is what I’m doing now, do shorter spurts and I’m getting stronger.*
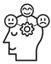	Behavior and Barriers	Most participants tried to continue their activities during treatment, but they were daily tasks instead of intentional exercises.	*We have never had a regular exercise program. Farming is strenuous and a lot of activity so we have never sat down and set up anything like a structured exercise program.*
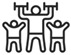	Exercising with a Partner	Most participants expressed a strong preference and mentioned that a partner would help with accountability and consistency.	*I absolutely have an accountability partner. I definitely need that. I do rely on support and I have a best friend who is invested in this journey with me.*
Section 2: Cancer-Specific Exercise Mobile App Receptiveness
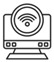	Internet Accessibility	Internet access is not a concern, and most participants are receptive to cancer-specific mobile apps.	*Yes, I would use an app and yes, I do have Internet at home.*
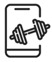	Exercising with a Partner and an app	Some participants noted that exercising with partners at different stages of life or health can support their routines. Some others find it more challenging than helpful.	*Yes, and no. We are in different stages of life. She’s probably in better shape than I am. It would depend on the situation if we have the same capabilities and restrictions taken into consideration.*
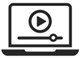	Role of Videos	The idea of videos was well-received by all participants.	*I think it would make it better. You can see what you are supposed to do.*
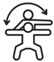	Role of a Trainer	The idea of a live exercise trainer was well-received by all participants.	*It would be nice to have a live person to know whether I’m doing the exercise right or not and they can go ahead and correct you on it.*
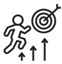	Motivation	Regaining pre-diagnosis strengths and abilities are key factors driving most participants’ motivation.	*I would say if you were exercising, hopefully would make a difference on you and your strength. If I could tell that I was getting stronger, it would be awesome.*

## Data Availability

The data analyzed during the current study are available from the corresponding author upon reasonable request.

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
