# Peer review of "Patient-Reported Perception of Exercise and Receptiveness to Mobile Technology in Cancer Survivors Living in Rural and Remote Areas"

_curroncol, 2025, doi:10.3390/curroncol32020067_

Round 1
Reviewer 1 Report
Comments and Suggestions for Authors
This manuscript reports a qualitative analysis of cancer survivors/care-givers’ perception of exercise and receptiveness to mobile technology in rural and remote areas. In my opinion, the paper’s goal is specific and provides some good insights of the concerned topic. However, there are also critical weaknesses too. Below are my comments and questions for the authors to consider in their efforts of improving the manuscript.
1. The title is clear, the results are well-presented, but the purpose (lines 97-98) missed one important part of the analysis: Receptiveness to a personalized cancer-specific exercise mobile app.
2. Section 2 “Patients and Methods” should be changed to “Sample and Methods”. Why? Participants are not only patients but also care givers. And sociodemographic information on participants is presented later under Section 3 “Results”.
3. Provide more sociodemographic information on the participants such as age, gender, race/ethnicity, educational level if they are available.
4. More direct quotes from the interviews/focus groups discussion could be provided under the Results.
5. The paper mentioned about “an exercise mobile application which can be individualized” but it seems this app was left imaginable to the participants. This is the most critical part of the discussion about whether and how patients would be ready to use such an app. There are many apps for this purpose with such a variation in the design and usability. How such an app would be individualized would have significant implications for participants to accept, use, and stay on. The study group seemed to make this app very generic to the participants to imagine, thus limiting their understanding and readiness to adopt such an app. Line 260: “Participants shared their thoughts on using personalized videos to help them engage with and stay committed to an app-based exercise program.”
There is no clear information if there was any app demonstration during the interview/focus group discussion. Did the researchers present a personalized cancer-specific exercise mobile app to the group? What app was that? Would it be freely downloaded from a popular source? How would such an app be individualized?
Author Response
Comments 1: The title is clear, and the results are well-presented, but the purpose (lines 97-98) missed one important part of the analysis: Receptiveness to a personalized cancer-specific exercise mobile app.
Response 1: Thank you for pointing this out. We agree with this comment. The purpose of this study has been edited according to this study's main focus (page 3, lines 117-121).
Comments 2: "Patients and Methods" should be changed to "Sample and Methods". Why? Participants are not only patients but also caregivers. And sociodemographic information on participants is presented later under Section 3 "Results".
Response 2: We agree with this comment. The title of Section 2 has been changed to "Sample and Methods" (line 122).
Comments 3: Provide more sociodemographic information on the participants such as age, gender, race/ethnicity, educational level if they are available.
Response 3: We agree with this comment. Therefore, we did not collect age, gender, race/ethnicity, and educational level. However, a demographic table has been revised to provide more sociodemographic information about the participants (Table 1), and a sentence related to the type of cancers in the narrative section has been removed and referred to in Table 1 (page 5, line 192).
Comments 4: More direct quotes from the interviews/focus groups discussion could be provided under the Results.
Response 4: We agree with this comment. Table 2 is a summary of the interviews/focus group discussion that has been added to provide more direct quotes from the participants (page 8, line 339).
Comments 5: The paper mentioned about "an exercise mobile application which can be individualized" but it seems this app was left imaginable to the participants. This is the most critical part of the discussion about whether and how patients would be ready to use such an app. There are many apps for this purpose with such a variation in the design and usability. How such an app would be individualized would have significant implications for participants to accept, use, and stay on. The study group seemed to make this app very generic to the participants to imagine, thus limiting their understanding and readiness to adopt such an app. Line 260: "Participants shared their thoughts on using personalized videos to help them engage with and stay committed to an app-based exercise program."
There is no clear information if there was any app demonstration during the interview/focus group discussion. Did the researchers present a personalized cancer-specific exercise mobile app to the group? What app was that? Would it be freely downloaded from a popular source? How would such an app be individualized?
Response 5: Thank you for pointing this out. The Introduction section (lines 110-117) includes a description of the cancer-specific exercise app to provide more context. However, our decision to present the exercise mobile application in a more general and conceptual manner was intentional. We wanted to avoid influencing participants' responses based on the app's current state. This approach allowed us to capture participants' preferences, needs, and innovative ideas without limiting feedback to the existing design. Moving forward, we will focus on incorporating these participant-driven preferences into the app to enhance its personalization, usability, and accessibility for individuals with cancer. This iterative process will help us ensure that the app meets user expectations and supports long-term engagement and adherence.
Reviewer 2 Report
Comments and Suggestions for Authors
Dear authors,
Your paper is very interesting and presents a significant problem related to the quality of life of oncology patients.
Minor revisions of work are required.
In the introduction, it is necessary to expand the section that refers to the possibility of using telemedicine interventions in oncology patients - whether they are available to them and whether they use them.
State how accessible applications related to physical activity are for different groups of patients, i.e. for oncology patients.
You mention solid tumors as one of the eligible criteria, and in the description of the sample you mention that patients with leukemia and multiple myeloma participated. Please check the eligible criteria. In the description of the sample, please indicate the age of the included participants.
One of the limitations also is the small number of respondents from each country.
Author Response
Comments 1: In the introduction, it is necessary to expand the section that refers to the possibility of using telemedicine interventions in oncology patients - whether they are available to them and whether they use them.
Response 1: We agree with this comment. The Introduction section (lines 96-109) has been updated to provide a broader context mainly focused on oncology patients and accessibility.
Comments 2: State how accessible applications related to physical activity are for different groups of patients, i.e. for oncology patients.
Response 2: We agree with this comment as well and the Introduction section (lines 98-99) has been updated to provide a broader context mainly focused on different groups of patients.
Comments 3: You mention solid tumors as one of the eligible criteria, and in the description of the sample you mention that patients with leukemia and multiple myeloma participated. Please check the eligible criteria. In the description of the sample, please indicate the age of the included participants.
Response 3: Thank you for catching it. The "solid tumor" eligibility criteria have been removed. However, the study group was not informed of the participant's age (line 132).
Comments 4: One of the limitations also is the small number of respondents from each country.
Response 4: We agree that it could be a limitation for some study settings. However, our focus group study aims to facilitate rich, detailed discussions that provide nuanced perspectives and generate qualitative data rather than aiming for statistical generalizability. Given this context, the number of participants was appropriate for achieving the study's aim of exploring perceptions and feedback in depth.
Round 2
Reviewer 1 Report
Comments and Suggestions for Authors
My previous comment: “The paper mentioned about “an exercise mobile application which can be individualized” but it seems this app was left imaginable to the participants. This is the most critical part of the discussion about whether and how patients would be ready to use such an app...”
Authors' response: “Our decision to present the exercise mobile application in a more general and conceptual manner was intentional. We wanted to avoid influencing participants' responses based on the app's current state. This approach allowed us to capture...”
My further comment: Then you want to clearly explain under the Methods section that you did NOT present any specific app to the participants and provided your rationale for not doing so.
Author Response
Comments 1: My previous comment: “The paper mentioned about “an exercise mobile application which can be individualized” but it seems this app was left imaginable to the participants. This is the most critical part of the discussion about whether and how patients would be ready to use such an app...”
Authors' response: “Our decision to present the exercise mobile application in a more general and conceptual manner was intentional. We wanted to avoid influencing participants' responses based on the app's current state. This approach allowed us to capture...”
Then you want to clearly explain under the Methods section that you did NOT present any specific app to the participants and provided your rationale for not doing so.
Response 1: Thank you for pointing this out. We agree that the rationale for not presenting the cancer-specific app to the participant was unclear. We provided the necessary clarification in the Methods section, page 4, lines 148-160. The rationale was highlighted in the text.
Additional changes: To improve the conclusion and provide a forward-looking perspective on the mobile app that has been discussed, we have added the following sentence on page 11, lines 407-409. "Future research should focus on testing cancer-specific exercise mobile apps to evaluate their feasibility, effectiveness, and potential to improve health outcomes for cancer survivors, particularly in rural and underserved areas."